# UFR2709, an Antagonist of Nicotinic Acetylcholine Receptors, Delays the Acquisition and Reduces Long-Term Ethanol Intake in Alcohol-Preferring UChB Bibulous Rats

**DOI:** 10.3390/biomedicines10071482

**Published:** 2022-06-22

**Authors:** Gabriel Gálvez, Juan Pablo González-Gutiérrez, Martín Hödar-Salazar, Ramón Sotomayor-Zárate, María Elena Quintanilla, María Elena Quilaqueo, Mario Rivera-Meza, Patricio Iturriaga-Vásquez

**Affiliations:** 1Laboratory of Experimental Pharmacology, Faculty of Chemical Sciences and Pharmacy, University of Chile, Santiago 8380494, Chile; mj_galvez@hotmail.com (G.G.); maria.quilaqueo@ug.uchile.cl (M.E.Q.); 2Instituto de Ciencias Químicas Aplicadas, Facultad de Ingeniería, Universidad Autónoma de Chile, Talca 3467987, Chile; juan.gonzalez01@uautonoma.cl; 3Programa de Doctorado en Ciencias, Mención Biología Celular y Molecular Aplicada, Universidad de La Frontera, Temuco 4811230, Chile; martinhodar@gmail.com; 4Laboratorio de Farmacología Molecular y Química Medicinal, Facultad de Ingeniería y Ciencias, Universidad de La frontera, Temuco 4811230, Chile; 5Centro de Neurobiología y Fisiopatología Integrativa (CENFI), Instituto de Fisiología, Facultad de Ciencias, Universidad de Valparaíso, Valparaíso 2360102, Chile; ramon.sotomayor@uv.cl; 6Program of Molecular and Clinical Pharmacology, ICBM, Faculty of Medicine, University of Chile, Santiago 8380494, Chile; equintanilla@med.uchile.cl

**Keywords:** UFR2709, voluntary ethanol intake, UChB rats, nicotinic antagonist

## Abstract

Alcoholism is a worldwide public health problem with high economic cost and which affects health and social behavior. It is estimated that alcoholism kills 3 million people globally, while in Chile it is responsible for around 9 thousand deaths per year. Nicotinic acetylcholine receptors (nAChRs) are ligand-gated ion channels expressed in the central nervous system, and they were suggested to modulate the ethanol mechanism involved in abuse and dependence. Previous work demonstrated a short-term treatment with UFR2709, a nAChRs antagonist, which reduced ethanol intake using a two-bottle free-choice paradigm in University of Chile bibulous (UChB) rats. Here, we present evidence of the UFR2709 efficacy in reducing the acquisition and long-term ethanol consumption. Our results show that UFR2709 (2.5 mg/kg i.p.) reduces the seek behavior and ethanol intake, even when the drug administration was stopped, and induced a reduction in the overall ethanol intake by around 55%. Using naïve UChB bibulous rats, we demonstrate that UFR2709 could delay and reduce the genetically adaptive impulse to seek and drink ethanol and prevent its excessive intake.

## 1. Introduction

Alcoholism is a worldwide public health problem with high economic cost and which affects health and social behavior. It is estimated that alcoholism kills 3 million people globally, while in Chile it is responsible for around 9 thousand deaths per year. Genetic and environmental conditions can lead to uncontrolled consumption of ethanol, one of the most used substances of abuse in the world [1]. Alcohol and nicotine are commonly co-abused [2,3] and many people who are considered heavy drinkers also smoke tobacco [4]. Specific common genes are suggested to regulate the susceptibility to both alcohol and nicotine dependence [2,5,6,7,8,9]. In this regard, there is evidence that shows a direct or indirect interaction between the nicotinic acetylcholine receptor (nAChR) and ethanol [10,11,12]. In addition, nAChRs were identified in the mesocorticolimbic dopamine system, and their role in alcoholism was proposed [2,13,14,15]. It was demonstrated that different nicotinic ligands can reduce ethanol ingestion in animal models and humans. [16,17,18,19,20,21]. The reward system or mesocorticolimbic system projects dopamine neurons from the ventral tegmental area (VTA) to the nucleus accumbens (NAc) and prefrontal cortex (PFC) [22,23]. This system is regulated by several neurotransmitters such as serotonin, glutamate, and acetylcholine, among others, which stimulate the firing rate of VTA DA neurons [24,25]. The activation of the reward system increases the NAc DA release which is mediated by reinforcement produced by drugs of abuse, such as amphetamine, cocaine, nicotine, and ethanol [26]. Although many studies about the rewarding properties of ethanol have been carried out [26,27], the elucidation of a specific mechanism for ethanol action is not completely resolved. Moreover, ethanol has been demonstrated to interact with glycine and GABAA receptors as a positive allosteric modulator and as a negative allosteric modulator of the N-methyl-D-aspartate (NMDA) receptor [28,29,30]. Additionally, it was shown that ethanol indirectly activates nAChR by increasing VTA acetylcholine levels [10,11,12,31], promoting the NAc DA release [32,33]. The activation of the reward system by nicotinic receptor agonism and its potentiation induced by ethanol [11,14,26,34] has opened a potential therapeutic target for treating nicotine addiction and alcoholism using nAChRs ligands.

nAChRs are ligand-gated ion channels that are present in the central nervous system (CNS), where the heteromeric α4β2 and the homomeric α7 subtypes are the most abundant [35,36]. However, different heteromeric nAChRs subtypes are expressed in the brain, i.e., α4β2α5, α6β2β3, α4β2α6, α4β4, α3β4, and α3β2, but they are less expressed or have a more specific distribution [35]. Several works have shown that ethanol effects involve different nAChRs subtype activation [37,38,39]. The α7 [40] and the α3β4 nAChRs [41,42] were implicated in ethanol intake, and the sedative effects of ethanol were associated with receptors containing the α5 nAChR subunit [43]. However, due to the relevant role of the α4β2 nAChR subtype in the brain reward system, it is suggested that this receptor subtype plays a pivotal function in ethanol intake. In this regard, it was established that a cholinergic–dopaminergic reward axis exists [44], which is affected by ethanol [45]. Dopaminergic ligands (agonist, partial agonist, and antagonist) have shown the role of the dopaminergic system in ethanol intake. The dopamine D1 receptor partial agonist SFK 38393, and the D2 receptor agonist Bromocriptine reduce ethanol preference and intake [46,47], and the 7-OH-DPAT, a D2/D3 receptor agonist, increases ethanol intake at lower doses. However, the dopaminergic antagonists SCH 23990, raclopride, and risperidone do not affect ethanol intake and preference [46].

Pharmacological studies using different nAChRs ligands (partial agonist, noncompetitive and competitive antagonist) have established a key role of nAChR in the modulation of the rewarding effects of ethanol [48]. Cytisine and varenicline, two partial agonists of nAChRs [49,50,51,52], decrease ethanol intake in rodents after a single dose or short-term administration [53,54,55]. The inhibitory effects of varenicline and cytisine on alcohol intake in alcohol-preferring University of Chile Bibulous (UChB) rats were confirmed in our lab [17]. On the other hand, mecamylamine, a non-competitive antagonist of central and peripheral nAChR, decreases ethanol consumption in rats [12,56,57] and decreases ethanol-induced DA release in Nac [56,57,58]. Additionally, erysodine, a competitive antagonist of nAChR, was shown to noticeably decrease ethanol consumption in UChB rats [19]. Moreover, we recently demonstrated, using UChB rats, that UFR2709, a competitive nicotinic antagonist [59], reduces voluntary ethanol intake in a dose-dependent manner without affecting body weight or locomotor activity, and does not produce a DA release from the striatum under i.p. injection [20]. In this work, we studied the effect of UFR2709 administration in the acquisition and the maintenance of ethanol intake in UChB rats that were chronically exposed to ethanol consumption. We found that UFR2709 decreased alcohol consumption at both stages even after ceasing its administration to the animals.

## 2. Materials and Methods

### 2.1. Drugs and Drinking Solutions

UFR2709-HCL (M.W. 255.74 g/mol) was prepared in our lab as previously described [57]. The injection volume was calculated related to body weight to reach the applied dose of UFR2709-HCl. Absolute ethanol (Merck, Darmstadt, Germany) and tap water were used to prepare a 10% *v*/*v* of ethanol solution. This concentration was used based on previous studies using UChB rats [60,61].

### 2.2. Animals

The experiments were carried out in adult male Wistar-UChB rats (*n* = 14), a half (*n* = 7) for saline solution and the other half for UFR2709 administration. The UChB rat line has been bred for over 90 generations to ingest 10% *v*/*v* ethanol solutions in preference to water [60,61]. UChB rats are genetically adapted and are considered a high drinking ethanol model and are used to screen drugs to treat chronic ethanol intake [61]. UChB rats were individually housed with free access to food and water. Animals were maintained under temperature- and humidity-controlled conditions with a 12 h light-dark cycle (lights off at 19:00 h). The animal’s weight was around 240 to 280 g. Ethanol intake experiments were performed under ARRIVE guidelines [62], at the faculty of Medicine, University of Chile, and were approved by the Animal Experimentation Ethics Committee of the University of Chile.

### 2.3. Effect of UFR2709 (2.5 mg/kg) on Voluntary Ethanol Intake by UChB Rats

Two-bottle free-choice experiments [17,60,61,63] were used to test the effects of a fixed dose of UFR2709 (2.5 mg/kg) on the voluntary ethanol intake of UChB rats. UChB rats were individually exposed to continuous access (24 h/day) of ethanol 10% *v*/*v* or distilled water in a two-bottle free-choice protocol. To prevent a potential position preference, bottles were daily alternated. All rats used were naïve but came from UChB bibulous mothers. The rats were then randomly divided into two groups (*n* = 7 per group), and a daily i.p. injection of UFR2709-HCl (2.5 mg/kg) or saline (1 mL/kg) was administered for 7 days at 15:00 h in a two-cycle paradigm, at days 1 to 7 and days 63 to 69 of the experiment (Figure 1). On days 1–2, animals were injected with UFR2709 or vehicle but were not given alcohol access, with the purpose to achieve pharmacologically effective drug concentration in the rats, before starting ethanol exposure. On day 3, ethanol access was started under 24 h continuous access two-bottle free-choice paradigm and maintained until day 100.

The second round of UFR2709 or saline administration was verified between days 63 to 69 (Figure 1). Animals were allowed *ad libitum* access to food and water. Ethanol and water intake were recorded at 14:00 h each day and expressed as g ethanol/kg/day and g water/kg/day, respectively. The weight was recorded once a week until the end of the experiments.

### 2.4. Statistical Analysis

Differences in ethanol intake between UFR2709- and saline-treated animals were analyzed using two-way ANOVA with Bonferroni’s multiple comparison test (Figure 2A, Figure 3 and Figure 4A,B). An unpaired *t*-test was used to analyze the effect of saline or UFR2709 administration on average ethanol intake (Figure 2B). Data are expressed as mean ± SEM. Statistical analyses were performed using Graph Pad Prism 8.0 software (Graph Pad Software, San Diego, CA, USA), and the level of statistical significance was set at *p* < 0.05.

## 3. Results

To determine the effect of a fixed dose of UFR2709 on the acquisition and long-term maintenance of voluntary ethanol intake, alcohol-preferring UChB rats were given a free choice between ethanol (10% *v*/*v*) and water. The dose of UFR2709 (2.5 mg/kg for 7 days) was selected from our previous work [20], where we demonstrated that this dose showed higher efficacy in reducing ethanol intake in this animal model of alcoholism. The first cycle of UFR 2709 administration (days 1–7) resulted in a marked delay in the acquisition of ethanol intake compared to the saline-treated animals (Figure 2A). Upon stopping UFR2709 administration, animals increase their ethanol intake steadily but did not reach the level of ethanol consumption shown by the saline-treated group. The second round of UFR2709 administration (days 63–69) resulted in a pronounced reduction of ethanol intake by the animals. Interestingly, after UFR2709 discontinuation, the rats showed a less pronounced increase in ethanol intake in comparison to the first cycle of UFR administration. A two-way ANOVA with Bonferroni’s multiple comparison test between drug, saline and days showed that 2.5 mg/kg of UFR2709 significantly reduced ethanol intake (Figure 2A) compared with the saline group over all the time of the experiment (interaction [F _(99,1200)_ = 1.992, *p* < 0.0001]; days [F _(99,1200)_ = 5.171, *p* < 0.0001]; treatment [F _(1,1200)_ = 2028, *p* < 0.0001]). For the first stage of the experiment (days 3 to 62) an unpaired *t*-test indicates a significant reduction of ethanol intake by UFR2709 (−52.77%) ([F _(6,6)_ = 1.929, *p* < 0.0001]). Additionally, at the second stage of the experiment, UFR2709 administration induced a 57.22% of reduction in ethanol intake compared with the saline group ([F _(6,6)_ = 1.596, *p* < 0.0001]) (Figure 2B).

Additionally, UFR2709 treatment significantly decreased the preference of the animals for the ethanol solution in the first stage of the experiment (days 3 to 62) compared to the saline-treated animals. In the second period of the experiment (days 63–100), the preference of the animals remained lower (approximately 60%) in comparison to the saline group, which showed ethanol preference close to 80% (Figure 3) (interaction [F _(99,1200)_ = 1.442, *p* = 0.004]; days [F _(99,1200)_ = 5.969, *p* < 0.0001]; treatment [F _(1,1200)_ = 928.7, *p* < 0.0001]).

A two-way ANOVA with Bonferroni´s multiple comparison test between drug, saline and days showed that 2.5 mg/kg of UFR2709 produced significant differences on total fluid intake (Figure 4A) compared to the saline group (interaction [F _(99,1200)_ = 1.748, *p* < 0.0001]; days [F _(99,1200)_ = 6.938, *p* < 0.0001]; treatment [F _(1,1200)_ = 350.7, *p* < 0.0001]). In the first stage of the experiment, the UFR2709 treatment did affect the total fluid intake (days 3 to 62) compared to the saline-treated animals. However, in the second period of the experiment (days 63–100), the total fluid intake of the animals treated with UFR2702 showed a lower total fluid intake in comparison to the saline group (Figure 4A). The administration of UFR2709 and the time course of the experiment did not affect the body weight compared to the saline group, and a normal increase in body weight was observed during the course of this study (interaction [F _(13,168)_ = 0.4239, *p* = 0.9596]; days [F _(13,168)_ = 21.12, *p* < 0.0001]; treatment [F _(13,168)_ = 32.55, *p* < 0.0001]) (Figure 4B).

## 4. Discussion

In this work, we studied the effects of a competitive nicotinic antagonist (UFR2709) on the acquisition and long-term ethanol consumption of alcohol-drinking UChB rats. This rat line has been bred for over 90 generations to ingest 10% *v*/*v* ethanol solution in preference to water [60,61] and could be considered a suitable model for screening drugs to treat chronic ethanol consumption [61]. We previously demonstrated that UFR2709 decreases ethanol intake at different doses in a short-term experiment (17 days), where we found that 2.5 mg/kg was the most effective dose of UFR2709 in reducing ethanol intake in UChB rats [20]. Therefore, we used this dose concentration to perform a long-term experiment.

One of the main differences between the protocol used in this experiment compared with our previous report is the fact that our rats, despite coming from bibulous mothers and fathers, were naïve to ethanol, and the acquisition of ethanol intake starts on day 3 after the drug administration. Previously, we used rats that were subjected to a homecage with free access to ethanol and after twenty days of ethanol intake, a stable plateau of ingestion was reached and the acquisition of ethanol intake behavior was established and this animal could be considered in a bibulous stage. Here, our protocol trends unveil that blocking the nicotinic systems associated with ethanol consumption before exposing rats to free access to ethanol, could decrease the instinct of the UChB rats to seek and ingest ethanol, considering that UChB rats have been genetically adapted for generations for the excessive consumption of ethanol. In this regard, rats were administered with the drugs two days before the start of ethanol access, only on the 3rd day were animals allowed ethanol consumption and water in the two-bottle free-choice paradigm, and the volumes of water and ethanol were recorded each day. Our results showed an interesting decrease in ethanol consumption mediated by UFR2709 administration compared to the saline-treated group. During the days of drug administration (days 1–7), animals reduce their impulsive behavior to ethanol intake significantly and when the drug administration was stopped, they progressively start to increase their ethanol ingestion but never reach the intake shown by the saline group. After the second period of UFR2709 administration, around the 60th day of the experiment, animals reduce their ethanol consumption to the levels observed after the first round of administration (~2.5 g ethanol/kg/day). When UFR2709 administration was stopped, animals slowly increase their ethanol intake but achieved a level of ethanol intake that was markedly lower than the level observed in vehicle-treated animals (~4.0 vs. 8.0 g ethanol/mg/kg), and the maximum level of ethanol consumption after the first round of UFR2709 (~6.0 g ethanol/kg/day) (Figure 2A). Our results indicate that UFR2709 significantly reduces the ingestion of ethanol even when the drug administration was stopped and produces a reduction of this behavior over all the time of the experiment. Additionally, we show that the body weight increases as rats become older, and no negative change was observed throughout the experiment at around 14 weeks (Figure 4B). On average, the total ethanol consumption of UChB rats treated with 2.5 mg/kg of UFR2709 was around 55% less compared with the saline group after two sessions of drug injection (Figure 2B) and a concomitant increase in water intake compared with the saline group. Our previous work shows that the effect of UFR2709 on ethanol intake does not result in inhibition of locomotor activity and no change in body weight was observed. Additionally, we demonstrated by microdialysis in the striatum that UFR2709 does not produce DA release, even more, it is possible to observe a small decrease in the DA levels [20]. Our study does not demonstrate the molecular mechanism involved in the reduction of ethanol consumption mediated by UFR2709 but clearly shows that nicotinic antagonism delays the acquisition and reduces the ethanol intake in long-term experiments and could be used as therapeutic drugs to treat or prevent alcohol abuse.

## 5. Conclusions

Our findings show that UFR2709 decreases ethanol consumption in UChB bibulous rats, suggesting that blocking brain nicotinic receptors inhibits the behavior of seeking and ingesting ethanol. The UChB rat line is genetically conditioned to ingest ethanol in preference to water. UFR2709 administered before ethanol access shows that this drug could stop the impulsive/genetically adapted necessity of ethanol ingestion. These findings further support the view that nicotinic receptors are implicated in the regulation of alcohol intake and that blocking these receptors reduces the instinctive necessity of ethanol intake. Further studies are needed to understand the molecular and biochemical mechanism involved in alcoholism mediated by nicotinic receptor antagonists.

## Figures and Tables

**Figure 1 biomedicines-10-01482-f001:**
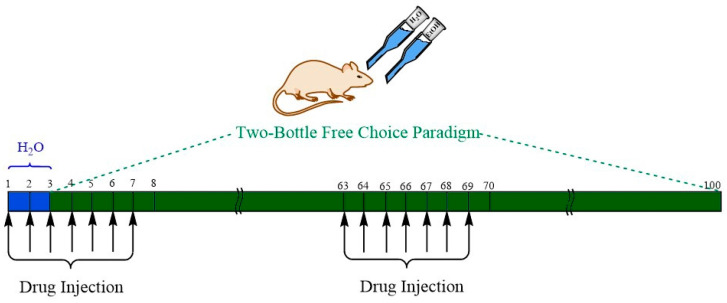
Arrows represent the days where UFR2709 and saline were administered, the blue square indicated the days where the drug was administered without access to ethanol, and the green timeline indicates the days where the ethanol intake protocol was conducted.

**Figure 2 biomedicines-10-01482-f002:**
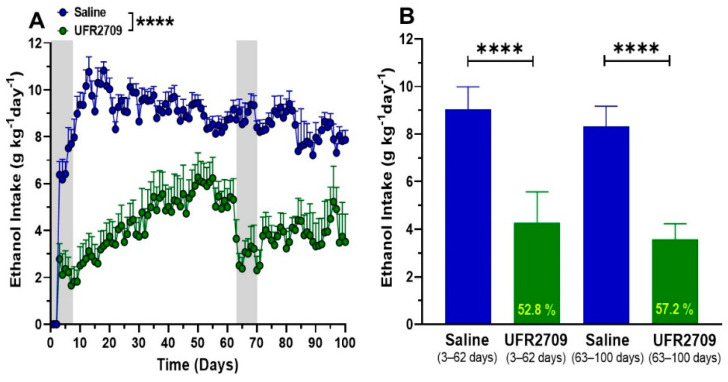
(**A**) A two-bottle free-choice experiment was used to test the effects of a fixed dose of UFR2709 (2.5 mg/kg) on the voluntary ethanol intake of UChB rats per 100 days. Each point represents the average ethanol intake for 7 animals per group during the experiment period. Rats (*n* = 7 per group) were administered a single i.p. injection of UFR2709 (2.5 mg/kg/day) or saline (1 mL/kg/day) at 15:00 h, and ethanol consumption was recorded at 14:00 h the next day. Drug administration was made in two stages, the first treatment was the days 1st to 7th and the second treatment was the days 63rd to 69th (labeled in gray in the graph). (**B**) Total average ethanol intake of the two stages of the experiments, days 3–62 in the first period and days 63–100 in the second one. An unpaired *t*-test was used to compare UFR2709 and saline group averages. Data are expressed as mean ± SEM (g/kg/day), and the number inside the bar indicates the percentage of reduction in ethanol intake compared to the saline-treated group. ********
*p* < 0.0001.

**Figure 3 biomedicines-10-01482-f003:**
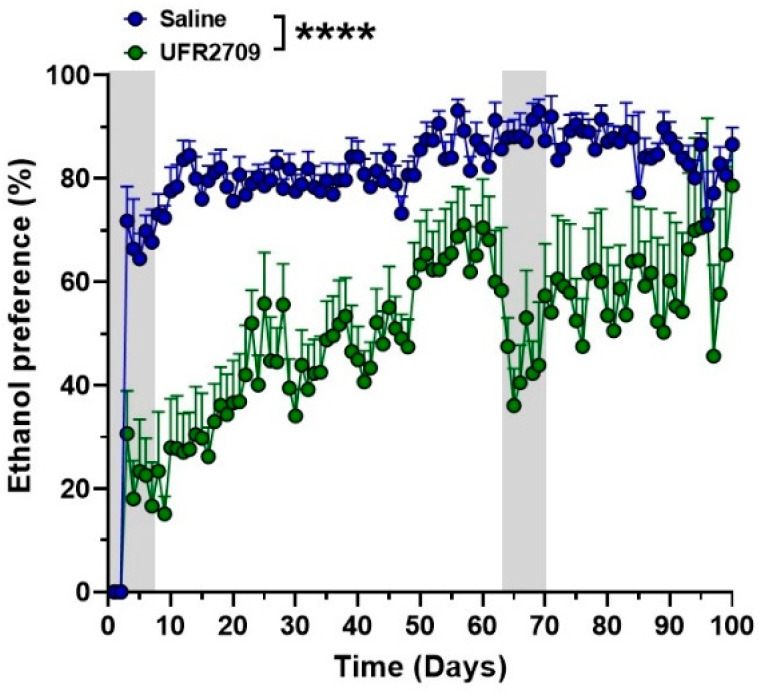
Preference for ethanol solution of the animals treated with UFR2709 or saline (gray bars). Ethanol preference data are expressed as mean ± SEM (g/kg/day) and represent the percentage of total daily fluid intake that was ingested from the bottle containing the ethanol solution. Two-way ANOVA with Bonferroni’s multiple comparison test was used to analyze the effect of UFR2709 treatment on ethanol preference. ********
*p* < 0.0001.

**Figure 4 biomedicines-10-01482-f004:**
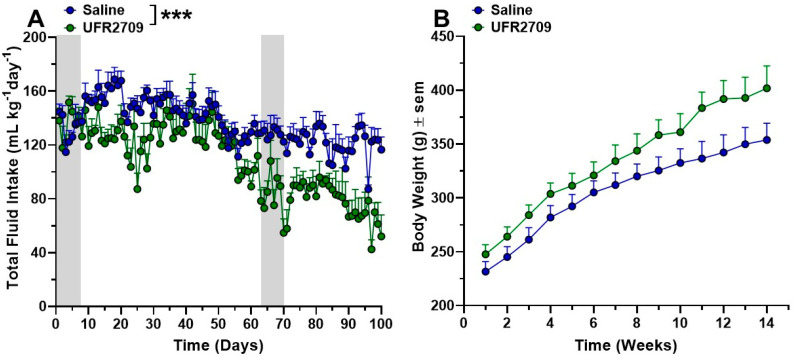
(**A**) Total fluid intake of 2.5 mg/kg of UFR2709 compared to the saline group. A two-way ANOVA with Bonferroni´s multiple comparison test between drug, saline, and days showed that UFR2709 produces significant differences in total fluid intake. (**B**) Data show the body weight of animal during the time of the experiment, all animal increases their body weight over the course of the experiment. Unpaired *t*-test was used to analyze the effect of UFR2709 treatment versus saline-treated group on the average ethanol consumption. *******
*p* < 0.001.

## Data Availability

Data are contained within the article.

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
