# Peer review of "UFR2709, an Antagonist of Nicotinic Acetylcholine Receptors, Delays the Acquisition and Reduces Long-Term Ethanol Intake in Alcohol-Preferring UChB Bibulous Rats"

_biomedicines, 2022, doi:10.3390/biomedicines10071482_

Round 1

Reviewer 1 Report

The manuscript “UFR2709, an antagonist of nicotinic acetylcholine receptors, reduces the acquisition and long-term ethanol intake in alcohol preferring UChB bibulous rats” by Galvez et al is a research article which describes the effects of UFR2709 on acquisition and long-term alcohol consumptions. The authors found that UFR2709 significantly reduced the alcohol intake behaviors in UChB rats. Therefore, the arthors suggest that UFR2709 could be used as therapeutics drugs to treat or prevents alcohol abuse. Generally, the subject is of interest and scientifically sound and contains essential issues. This topic is also of importance for treatment of alcohol abuse. The manuscript has been well organized and written. However, I have some concerns on the paper.

1.      In abstract, UChB should be spelled out. Or the authors should add the explanation of it.

2.      Please add the explanation why the authors used only male rats in this study!

3.      In the discussion, the authors described cholinergic-dopaminergic reward axis is involved in alcohol consumption. In this regard, it is interesting to test how the dopamine agonist or antagonist affects the effects of UChB on alcohol consumption in UChB rats.

Author Response

Reviewer 1:

The manuscript “UFR2709, an antagonist of nicotinic acetylcholine receptors, reduces the acquisition and long-term ethanol intake in alcohol preferring UChB bibulous rats” by Galvez et al is a research article which describes the effects of UFR2709 on acquisition and long-term alcohol consumptions. The authors found that UFR2709 significantly reduced the alcohol intake behaviors in UChB rats. Therefore, the authors suggest that UFR2709 could be used as therapeutics drugs to treat or prevents alcohol abuse. Generally, the subject is of interest and scientifically sound and contains essential issues. This topic is also of importance for treatment of alcohol abuse. The manuscript has been well organized and written. However, I have some concerns on the paper.

  1. In abstract, UChB should be spelled out. Or the authors should add the explanation of it.

UChB was changed by “University of Chile bibulous (UChB) rats”

  1. Please add the explanation why the authors used only male rats in this study!

Male rats have been used for behavior studies for a long time due to they have not hormone cycles alteration but we understand that at this time, it is accepted the use of male and female animals for behavioral experiments and we going to conduct our next experiments on this direction.

  1. In the discussion, the authors described cholinergic-dopaminergic reward axis is involved in alcohol consumption. In this regard, it is interesting to test how the dopamine agonist or antagonist affects the effects of UChB on alcohol consumption in UChB rats.

We added a new paragraph indicating the role of dopamine receptor ligands in EtOH intake. Lines 75-80

“In this regard, it has been established that exists a cholinergic-dopaminergic reward axis [44], which is affected by ethanol [45]. Dopaminergic ligands (agonist, partial agonist, and antagonist) have shown the role of dopaminergic system in ethanol intake. The dopamine D1 receptor partial agonist SFK 38393, and the D2 receptor agonist Bromocriptine reduce ethanol preference and intake [46, 47], 7-OH-DPAT a D2/D3 receptor agonist increase ethanol intake at lower doses. However, the dopaminergic antagonist SCH 23990, raclopride, and risperidone do not affect ethanol intake and preference [46]”.

Reviewer 2 Report

Compared to previously Quiroz et al. (2019, doi: 10.3389/fphar.2019.01429), this paper investigated alcohol consumption and preference after ethanol intake of the nicotinic acetylcholine receptors antagonist, UFR2709, to UChB rats for 100 days.

Only two studies related to UFR2709 were identified in PubMed, one in Zebrafish and the other by the authors of this paper. Therefore, I think that research related to UFR2709 is just the beginning. I commend the researchers for carrying out the experiment for a long time, the previous study showed that the rats were ingested with alcohol for 20 days, and this study showed the results after 100 days of ingestion.

The resulting presentation was similar to that of previous papers. Therefore, it would be desirable to describe in detail the differences and importance compared with previous studies.

Also, I think the results shown in this study are too few with three figures. By showing only pictures of alcohol consumption and preference, I do not know the exact mechanism of why such a result was derived. Therefore, I think that it will be a more accurate and high-quality thesis if we show the verification results for the cytological, histological, molecular, or biochemical mechanisms related to the Nicotinic Acetylcholine Receptor. Or, if it was compared with other proven competitive Nicotinic Acetylcholine Receptor Antagonists, it would have led to better results as claimed by the researchers.

I'm not sure what you mean by "the acquisition" in the title in relation to the results. You might want to consider the title.

English proofreading is required throughout the paper.

In Abstract, it is as follows.

Line 20: and affecting --> and affects

Lines 21-22: globally while in Chile it is responsible of around 9 thousand deaths per year.--> globally, while it is responsible for around 9 thousand deaths per year in Chile. 

Line 23: system and they --> system, and they

Line 24: suggested to modulate --> suggested modulating 

Lines 24-25: In a previous work, we have demonstrated that a short-term --> Previous work demonstrated that short-term 

Line 28: UFR2709 (2.5 mg/kg i.p.) reduce --> UFR2709 (2.5 mg/kg i.p.) reduces

Line 30: Using naïve UChB bibulous rats, we 

Author Response

Reviewer 2:

Compared to previously Quiroz et al. (2019, doi: 10.3389/fphar.2019.01429), this paper investigated alcohol consumption and preference after ethanol intake of the nicotinic acetylcholine receptors antagonist, UFR2709, to UChB rats for 100 days.

Only two studies related to UFR2709 were identified in PubMed, one in Zebrafish and the other by the authors of this paper. Therefore, I think that research related to UFR2709 is just the beginning. I commend the researchers for carrying out the experiment for a long time, the previous study showed that the rats were ingested with alcohol for 20 days, and this study showed the results after 100 days of ingestion.

Many thanks for your comment, and you are right, UFR2709 was published in 2013 as a nicotinic antagonist, named compound 1 in the paper. UFR along with others were characterized by their affinity (binding exp), and functional (antagonist activity) (Faundez-Parraguez et al, 2013). But, around 2015/2016 we start to use zebrafish as a model for addiction and we test the UFR compound, the paper that you mention. At that time, we meet with Dr. Mario Rivera from the University of Chile that have access to UChB rats, and we write the second paper, and we continue working where as a result we write this ms. At the moment, two Ph.D. students are doing their thesis on rats and zebrafish looking at the behavior and the molecular and biochemical mechanism involved in the effects of UFR in nicotine and ethanol addiction.

The resulting presentation was similar to that of previous papers. Therefore, it would be desirable to describe in detail the differences and importance compared with previous studies.

In this work, we study the effect of UFR2709 on the acquisition of ethanol intake behavior, and in the previous work, we test the effect of UFR2709 over the chronic ingestion of ethanol, where the acquisition has been established during 20 days of free access to ethanol before administering the drug. The effect on the chronic intake was made after 60 days of continuous access to ethanol, in this work. The administration was reduced to 7 days to minimize the tolerance to the drug, whereas previously the treatment was for 17 days. The drug was administered two times, which shows to be additive.

The discussion was modified in the direction of our commentary above.

Also, I think the results shown in this study are too few with three figures. By showing only pictures of alcohol consumption and preference.

This work has been submitted as a short communication. So, we show few figures but with enough information about our findings. But, if the reviewer considers relevant to include water intake, we can do it, but we think that the total volume of liquids considers this analysis.

I do not know the exact mechanism of why such a result was derived. Therefore, I think that it will be a more accurate and high-quality thesis if we show the verification results for the cytological, histological, molecular, or biochemical mechanisms related to the Nicotinic Acetylcholine Receptor. Or, if it was compared with other proven competitive Nicotinic Acetylcholine Receptor Antagonists, it would have led to better results as claimed by the researchers.

As we explain above, UFR along with others were characterized by their affinity (binding exp), and functional (antagonist activity) (Faundez-Parraguez et al, 2013). But, around 2015/2016 we start to use zebrafish as a model for addiction and we test the UFR compound, the paper that you mention. At that time, we meet with Dr. Mario Rivera from the University of Chile that have access to UChB rats, and we write the second paper. At the moment, two Ph.D. students are doing their thesis on rats and zebrafish looking at the behavior and the molecular and biochemical mechanism involved in the effects of UFR in nicotine and ethanol addiction. Mecamylamine a non-competitive antagonist, and erysodine a competitive antagonist has been described as reducing ethanol intake, and are discussed at the end of the introduction. -Lines 87-91.

We added the following paragraph to the conclusion:

“Further studies are needed to understand the molecular and biochemical mechanism involved in alcoholism mediated by nicotinic receptors antagonist”.

I'm not sure what you mean by "the acquisition" in the title in relation to the results. You might want to consider the title.

This point was added and clarified in the manuscript, we think it is important to consider that the compound is effective both in individuals who have a genetic predisposition to drink ethanol but are naïve and in those who have already established chronic consumption. Using naïve UChB rats, we demonstrated that UFR2709 could delay the acquisition and decrease the intake in a long-term protocol. Line 226-230.

We modified the title on this way.

“UFR2709, an antagonist of nicotinic acetylcholine receptors, delays the acquisition and reduces long-term ethanol intake in alcohol-preferring UChB bibulous rats.”

English proofreading is required throughout the paper. English was modified and amended overall the ms.

Round 2

Reviewer 2 Report

I think the author has sufficiently answered the question in the review presented earlier. It has been revised a lot in the introduction, so it is somewhat more understandable, but I still think that the amount of results is small. As mentioned in the previous review, I think molecular biological or biochemical data are needed. I think that the fact that there are physiological changes is not enough to be satisfied.
